

# Evaluation of snow depth and snow-cover over the Tibetan Plateau in global reanalyses using in-situ and satellite remote sensing observations

5   Yvan Orsolini[1], Martin Wegmann[2], Emanuel Dutra[3], Boqi Liu[5], Gianpaolo Balsamo[4], Kun Yang[6,7], Patricia de Rosnay[4], Congwen Zhu[5], Wenli Wang[6,7], Retish Senan[4]

1 NILU - Norwegian Institute for Air Research (Norway)

2 Alfred-Wegener Institute (Germany), formerly at Institut des Geosciences de l'Environnement, University of
10  Grenoble, Grenoble, France

3 Instituto Dom Luiz (IDL), Faculdade de Ciências, Universidade de Lisboa, Portugal)

4 European Centre for Medium-Range Weather Forecasts (ECMWF) (United Kingdom)

5 Institute of Climate System, Chinese Academy of Meteorological Sciences, (China)

Department of Earth System Science, Tsinghua University (China)

7 Institute of Tibetan Plateau Research of Chinese Academy of Sciences (China)

*Correspondence to*: Yvan Orsolini (orsolini@nilu.no)

**Abstract.** The Tibetan Plateau (TP) region, often referred to as the Third Pole and, is the world highest plateau and exerts a considerable influence on regional and global climate. The state of the snowpack over the TP is a major research focus due to its great impacts on the headwaters of a dozen major Asian rivers. While many studies have attempted to validate
atmospheric re-analyses over the TP area in terms of temperature or precipitation, there have been –remarkably– no studies aimed at systematically comparing the snow depth or snow cover in global re-analyses with satellite and in-situ data. Yet, snow in re-analyses provides critical surface information for forecast systems from the medium to sub-seasonal time scales. Here, snow depth and snow cover from 5 recent global reanalysis products are inter-compared over the TP region, and evaluated against a set of 33 in-situ station observations, as well as against the Interactive Multi-sensor Snow and Ice
Mapping System (or IMS) snow cover and a satellite microwave snow depth dataset. The high temporal correlation coefficient (0.78) between the IMS snow cover and the in-situ observations provides confidence in the station data despite the relative paucity of in-situ measurement sites and the harsh operating conditions.



While several re-analyses show a systematic over-estimation of the snow depth or snow cover, the reanalyses that assimilate local in-situ observations or IMS snow-cover are better capable of representing the shallow, transient snowpack over the TP region. The later point is clearly demonstrated by examining the family of re-analyses from the European Centre for Medium-Range Weather Forecasts (ECMWF), of which only the older ERA-Interim assimilated IMS snow cover at high altitudes, while ERA5 did not consider IMS snow cover for high altitudes. One missing process in the re-analyses is the blown snow sublimation, which seems important in the dry, windy and cold conditions of the TP. By incorporating a simple parametrisation of this process in the ECMWF land re-analysis, the positive snow bias is somewhat alleviated. Future snow reanalyses that optimally combine the use of satellite snow cover and in-situ snow-depth observations over the Tibetan Plateau region in the assimilation and analysis cycles, along with improved representation of snow processes, have the potential to substantially improve weather and climate prediction and water resources applications.

## 1 Introduction

Often referred to as the Third Pole, the Tibetan Plateau (TP) region is the world highest plateau, with an average elevation of 4000 m above sea level. Due to its spatial extent, elevation and geographical position in the mid-latitudes, it exerts a considerable influence on regional and global climate. The formation and variability of the Asian summer monsoon in particular, is affected by the TP through thermal and mechanical effects (Wu et al., 2012, 2015; Xiao and Duan, 2016), with remote impacts both downstream (e.g., Zhang et al., 2004; Xue et al., 2018) and upstream ( Lu et al., 2017). In autumn or winter, snow anomalies over the TP have also been linked to wave trains extending downstream over East Asia and the Pacific (Liu et al., 2017; Li et al., 2018).

Given its importance for climate and given that climate change strongly affects the region (Yang et al., 2014), the cryosphere over the TP is closely monitored in terms of glacier shrinking (Yao et al., 2012; Treichler et al., 2018), lake expansion (Zhang et al., 2017; Treichler et al., 2018) or change in the snowpack (Chen et al., 2017; Wang et al., 2017; Wang et al., 2018).

The extent and variability of the snowpack over the TP has been a major focus of investigation because of the role of snow in the surface energy balance and in the hydrological cycle, as well as its potential impact on the large-scale circulation through radiative or thermodynamical feedbacks (Xiao and Duan, 2016; Lin et al., 2016; Henderson et al., 2018). In addition, water storage over the TP area affects the headwaters of a dozen major Asian rivers, affecting the livelihood of 22 % of the world population (Immerzeel et al., 2010).

Several studies have aimed at quantifying the seasonal, interannual and decadal variability of the TP snowpack (e.g., Basang et al., 2017 and references therein). The seasonal snow cover over the TP is unique compared to other mid-latitude regions or to higher latitudes, because of its location in the 27° N -40° N latitude belt, its high elevation and its distinct shallow, patchy and short-lived snowpack. In addition, the TP also receives a high amount of solar short-wave radiation, hence the snow albedo effect tends to be strong over that region. The TP is also a challenging region for snow-related research due to the complex terrain and the relative paucity of in-situ observation stations in this vast, sparsely populated expanse, especially in



its western part. There is, nevertheless, a substantial number of meteorological stations over the TP region, operated by the Chinese Meteorological Administration (CMA), which have been used in many climatological studies (e.g., for recent studies, Basang et al., 2017; 2018; Li et al., 2018). Since most of the stations are located in inhabited valleys, below 4000 m, in the southeast TP, the representativeness of this in-situ data for the TP as a whole is questionable.

Advances in satellite remote sensing have however provided invaluable information on the state of this high-elevation snowpack (Basang et al., 2017; 2018; Yang et al., 2015; Menegoz et al., 2013), often through blending data from multiple instruments to combine the high spatial resolution of optical data with the all-weather capability of microwave (MW) data. Since 2004, NOAA has provided an operational, daily, multi-sensor snow cover product (the Interactive Multisensor Snow and Ice Mapping System or IMS) at a high-resolution (4-km), by combining optical, infrared and MW satellite data and

station data (Helfrich et al., 2007). The accuracy of the IMS product compared to station data over the TP has been evaluated by Yang et al. (2015) and Li et al. (2018), who found the pixel matching accuracy (snow or no snow) to be over 90 %. Another widely used satellite optical product is the Moderate-resolution Imaging Spectroradiometer (MODIS) 8-day snow cover (Riggs and Hall, 2015; Basang et al., 2017). Joint analysis of MODIS and station data by Basang et al. (2017) indicated that, despite large snow cover fluctuations from day to day and the non-synchronicity of these two datasets, there is

a high temporal correlation (0.77) if the station data is pre-processed in 8-day bins in a similar fashion as the MODIS data (i.e., the station data is considered snow-covered in the 8-day period, if snow covered for at least one day). Nevertheless, Basang et al. (2017) also showed that the spatial distribution of snow cover is uneven: the snow cover fraction is less than 21 % for 70 % of the TP area, and yet it can be up to 40 % in the eastern part of the TP. They also indicated a discrepancy in the seasonality of the maximum snow cover between station data and MODIS: the former had higher snow cover in the spring,

when precipitation increases and falls as snow in the southeast TP. On the contrary, MODIS had a higher winter snow cover, more representative of the mean conditions over the plateau. These latter discrepancies point out at large spatial differences in the characteristics of precipitation. Snow depth products are also available from satellite remotely sensing. Snow depth retrieval from MW observations is difficult in regions with complex topography, and is also hampered by non-heterogeneity of snow grain size and of vegetation cover. Few MW snow depth products have been thoroughly evaluated over the TP, a

region characterised by a sparse and rapidly melting snowpack where, given the small snow grain size of fresh snow, the MW retrievals can lead to large errors (Dai et al., 2017). Nevertheless, a long-term MW snow depth product has been developed to account for the specificity of the TP by Che et al. (2008) and Dai et al. (2017).

Atmospheric re-analyses comprehensively assimilate in-situ, balloon-borne, aircraft and satellite observations into a forecast model, and form an essential tool in meteorological and climate research (e.g., Bronnimann et al. 2018; Fujiwara et al.,

2017). Furthermore, re-analyses serve as initial conditions for prediction models, from short-term to monthly and seasonal forecasts. A thorough up-to-date description and intercomparison of atmospheric re-analyses is found in Fujiwara et al. (2017). On the other hand, data assimilation in land surface models is performed separately from the atmospheric data assimilation due to the different nature of observations and methodologies involved (Hersbach et al., 2018). Early land re-analyses were performed as offline simulations without any actual assimilation of observations, but land data assimilation is





rapidly evolving. For example, the newest operational seasonal forecasting system 5 (known as SEAS5, Johnson et al. 2018) and the Integrated Forecast System (IFS) used for medium-range prediction at the European Centre for Medium-Range Weather Forecast (ECMWF), both rely on a land surface data assimilation system using a 2-dimensional optimal interpolation, which blends IMS snow cover and in-situ snow depth observations with the model background (de Rosnay et

al., 2014, 2015). This evolution is partly motivated by the renewed interest in tapping into the potential of the land surface, and snow in particular, to improve the prediction skill at the subseasonal-to-seasonal time scale (Orsolini et al., 2013). Concerning the TP region more specifically, Lin et al. (2016) assimilated snow cover fraction from MODIS and/or water storage from GRACE in a land model and showed improvement in seasonal atmospheric temperature prediction resulting from the improved land conditions. The impact of realistic land initialization over the TP region on the Indian Summer

Monsoon (ISM) has also been investigated (Senan et al., 2016; Halder et al., 2017; Rai et al., 2015). In particular, the impact of springtime initialisation on predicting the ISM onset was investigated by Senan et al. (2016) using the coupled ocean-atmosphere seasonal forecasting System 4 of the ECMWF. They found that, in years with anomalously high mean snow depth over the TP, as determined from the ECMWF land re-analyses, the ISM onset was observed to be delayed by about 8 days, leading to persisting dry and warm conditions over India.  Half of this delay could be attributed to the initialization of

snow over the TP region, highlighting the importance of land initialisation over that region for seasonal forecasting of the ISM onset.

While many studies have attempted to validate atmospheric re-analyses over the TP area in terms of temperature, precipitation or snowfall (Wang et al., 2012; Palazzi et al., 2013; Zou et al., 2014; Viste et al., 2015), there have been remarkably few studies aimed at evaluating and comparing the snow depth or snow cover in the re-analyses. Snow in long-

term climate reanalyses over the plains of central and northern Russia have been evaluated against station data, showing surprisingly good performance (Wegmann et al., 2017). However, the terrain there is not as complex as over the TP. Previous studies of snow evaluation over the TP have focused purely on the remote sensing snow products (Basang et al., 2017; Yang et al., 2015; Dai et al., 2017). We argue that evaluation of snow re-analyses is of interest per se, since these re-analyses provide initial, critical surface information for the forecast systems, and link surface conditions with atmospheric

dynamics.

The aim of this study is to inter-compare snow cover and snow depth over the TP region in a set of modern atmospheric and land re-analyses, and to evaluate them against in-situ snow observations and selected satellite-based remote sensing products. A second goal is also to better characterise the snow biases in the re-analyses and to identify their origin. Sensitivity experiments are then carried out with a land surface model to assess the potential relevance snow processes in

driving the identified biases.

## 2  Data and methods



Maps of our study area with the orography at the resolution of the atmospheric re-analyses are shown in Figure 1. We make use of 3 re-analyses produced by ECMWF, namely the latest generation of atmospheric re-analyses (ERA5) (Hersbach et al, 2018), its older counterpart (ERA-Interim, hereafter ERA-I) (Dee et al., 2011), as well a land re-analysis obtained by running the ECMWF land model (HTESSEL) off-line, forced by the ERA5 meteorological forcing. Here, we refer to the latter as

Offline-ERA5L-CRTL since it is not yet the officially released version of ERA5-Land expected to become available online in 2019 (https://climate.copernicus.eu/climate-reanalysis). In addition, we also make use of the Japanese Re-Analyses (JRA-55) (Kobayashi et al., 2015), and the NASA Modern-Era Retrospective analysis for Research and Applications, Version 2 (MERRA2) (Gelaro et al., 2017). We didn't carry out an exhaustive intercomparison of all existing global re-analyses, but we chose the aforementioned analyses because they belong to the latest generation (ERA5, MERRA2, JRA-55), or else

because they incorporate snow observations over the TP region (ERA-I, JRA-55). Some general characteristics of the re-analyses are shown in Table 1, and more detailed information about the treatment and assimilation of snow variables is provided in the Appendix. Further information about the re-analyses systems can be found in Fujiwara et al. (2017) and Wright et al. (2018).

For the evaluation of re-analyses, we use station data, the multi-sensor IMS snow cover product and satellite MW data, as

detailed below.

## 2.1 Station Data

We obtained five years (2009 to 2013) of in-situ daily meteorological observations from 33 stations over the TP from CMA. The observations comprise minimum and maximum temperature, snow depth and total precipitation. The measuring time is 00:00 UTC (0800 Beijing time). The operational procedures dictate that, when more than 50 % of the ground surrounding

the weather station is covered by snow, then the snow depth (SD) is measured. When converting in-situ snow depth to a snow cover fraction (SCF), a 100 % SCF is assumed when snow depth exceeds 1 cm. The in-situ observations also provide the snow cover days (SCD), i.e. the number of days when the snow covers more than 50 % of the ground in the sight from the station. The geographical locations of the stations are shown on Figure 1, along with their altitudes in comparison to the topography resolved by the four atmospheric re-analyses. Most of the stations are located in inhabited valleys, below 4000m,

in the southeastern part and are not representative of the TP region as a whole. There is only one station in the western part of the TP, west of 85° E. In-situ observations have several sources of uncertainty. Here, we highlight two sources: (i) the stations might not be fully representative of their local surroundings due to the complex nature of the terrain, and (ii) the quality of the records could be affected by the harsh operating conditions. For example, strong winds limit the instrument ability to record the amounts of falling snow or solid precipitation, a phenomenon called undercatch.


## 2.2 Interactive Multisensor Snow and Ice Mapping System (IMS)





We use SCF from IMS, a multi-instrument, near-real time daily product covering the Northern Hemisphere with a pixel resolution of 4 km. It uses time-sequenced imagery from geostationary satellites, as well as infrared (AVHRR) and MW (S SM/I) satellite observations, and station data. IMS provides a binary (1/0) snow cover information: either 1 if more than 50

% of the 4-km pixel is covered by snow, or 0 (snow free) otherwise. Since, in this paper, we compare IMS data to re-analyses, we aggregate the 4-km product to a 0.25-degree grid, comparable to highest horizontal resolution among the re-analyses. This is done by counting the number of pixels with a value of 1 in a gridbox, assuming they represent 100 % cover - this gives the "high estimate". If we assume that a value of 1 represents only 50 % of the 4 km pixel, we obtain the "low estimate", for which the maximum SCF possible is 50 %. These two estimates provide a range of uncertainty, consistent with

the IMS product generation.

The choice of IMS snow cover is motivated by its use in the ECMWF analysis system. A key point is that it is used differently in ERA-I (Drusch et al., 2004) than in ERA5 (de Rosnay et al., 2014, 2015; Hersbach et al., 2018). First, the high-resolution 4 km product is used in ERA5, while the coarser resolution 24 km-product is used in ERA-I. More importantly, IMS data is not used above 1500 m, i.e., in high altitude regions such as the TP, in ERA5 while it was still in

use in ERA-I. When comparing the snow cover in IMS with the one in ERA-I or ERA5, one has to recall that these are not independent datasets. Nevertheless, the different usage of IMS data between ERA5 and ERA-I allows to highlight the importance of snow cover analysis and assimilation over the TP. In that region, in-situ data available for the numerical weather prediction community for real time applications is still scarce.

**2.3 Microwave satellite data**

A long-term (1978-2010) MW daily snow depth product has been developed to account for the specificity of the TP by Che et al. (2008) and Dai et al. (2017) at the Cold and Arid Regions Environmental and Engineering Research Institute (CAREERI) of the Chinese Academy of Sciences. We used the two years (2009 and 2010) that overlap with our comparison period (2009-2013), the latter being dictated by our station data record. Over these two years, the MW product is based on

measurements from the NASA Advanced Microwave Scanning Radiometer for Earth Observing System (AMSR-E). The gridded dataset has a horizontal resolution of 0.25 by 0.25 degree.

# 3 Results

## 3.1. Snow depth

Figure 2 presents a comparison of daily SD over the 5-year period (2009-2013) between the observations, comprising both in-situ station and MW data, and the 5 re-analyses collocated at the station coordinates. The comparison is an average over the 33 stations. The in-situ observations, even in the mean over all available stations, reveal a very thin and rapidly





fluctuating snowpack, with episodes of fast build-up and decay, followed by periods void of snow, even in mid-winter. This concurs with previous studies showing that large parts of the TP can remain snow-free even in winter (Basang et al., 2017; Li et al., 2018). A quantitative estimate of the discrepancy between in-situ daily observations and the other datasets as a root-mean-square error (RMSE) is presented in Fig. 4 (left panel). The RMSE was calculated for the two years 2009 and 2010

when satellite MW data is available. We also calculated the temporal correlation matrix between the different SD datasets (Table 2), using daily data year-round over the 5-year period, except for the satellite MW data where only the two years 2009 and 2010 are used.  It is clearly apparent that the re-analyses show a regular seasonal cycle, with a snowpack that grows nearly steadily during the cold season and culminates in February or March. This is unlike the in-situ observations which show rapidly fluctuating snow increases on a near-zero accumulation throughout winter.  In comparison with in-situ

observations, JRA-55 has the best performance among re-analyses for both the RMSE and the temporal correlation. MERRA-2, ERA5 and Offline-ERA5L-CTRL re-analyses considerably over-estimate the seasonal maximum SD by a factor 5 to 10. MERRA-2 also has remaining SD in summer. Among the ECMWF family of re-analyses, the older ERA-I is performing best in terms of RMSE. Nevertheless, the temporal correlation is similar to the one obtained with ERA5, indicating that the newer, higher resolution ERA5 similarly captures the snow variability despite its large positive bias. The

MW satellite data is also overestimating SD, as noted by Yang et al. (2015) or Dai et al. (2017). Like the re-analyses, it shows a progressive snow accumulation throughout the cold season. The temporal correlation with the in-situ data is poorer than for the re-analyses though (0.32), but is established over two years only. The RMSE error is nevertheless comparable to ERA-I and the MW data is able to represent shallow layers of the order of 5 cm or less.

It is not surprising that JRA-55 performs best, since it assimilates SDs from a dozen CMA stations over the TP area and

from satellite MW data. Note that the assimilated MW data is not processed by the same algorithm as the CAREERI MW data used here. A key factor in the relatively good performance of ERA-I is the fact that it assimilates SCF from IMS, even in high altitude regions, hence also over the TP. On the contrary, as mentioned earlier, assimilation of IMS SCF above 1500m was discontinued in the production of ERA5. The tendency to reduce or remove the snowpack provided by the IMS observational constraint during assimilation appears to play a major role in bringing ERA-I SDs significantly closer to the in-

situ observations.

### 3.2. Snow cover

Figure 3 presents a comparison of daily SCF over the 5-year period (2009-2013) between the in-situ station observation and the IMS satellite data and the 5 re-analyses, collocated at the station coordinates. Again, an average over the 33 stations is presented. The IMS data is represented by the range between the low and high estimates (see Section 2). The correlation

matrix and the RMSE for SCF are provided in Table 2 and in Fig. 4 (right panel) respectively, calculated over the 5-year period. There is good agreement between in-situ observations and the IMS low estimate: the station-mean year-round correlation (Table 2) over the 5 years (2009-2013) is 0.78. In Li et al. (2018), the correlations between SCF at 55 CMA stations and IMS SCF during the 10 winters between 2000 and 2010 ranged from 0.39 to 0.79, with a 10-winter average of 0.56. The consistency of the temporal correlation with Li et al. (2018) indicates that the satellite snow cover data readily



captures the rapidly fluctuating snowy events. It also provides some confidence in the station data despite the harsh operating conditions for in-situ measurements, and the spatial degradation applied to IMS (from 4 km to 0.25 degree).

It is clearly apparent that ERA5 and Offline-ERA5L-CTRL re-analyses again considerably overestimate SCF. JRA-55 is worse than MERRA-2 for SCF, contrarily to the case of SD. As described in the Appendix, the a-posteriori conversion from

SD to SCF varies among re-analyses. For JRA-55, a thin 2 cm layer is equivalent to 100 % SCF, hence JRA SCF is persistently high. While the in-situ station data (e.g., Fig. 3) also displays thin layers of a few cm, they are very transient.

Further validation is provided in terms of the annual-mean SCDs. Table 3 compares SCDs among the datasets, both for the 5-year average and for individual years, based on monthly-mean SCD values. The values of SCDs compare well between in-situ observation and IMS. On the other hand, ERA5 and JRA55 re-analyses largely overestimate SCDs (ERA5 by nearly

three times), while ERA-I is closer to the observed values. The temporal correlation coefficients for monthly SCDs tend to be higher than that for the daily SCF or SD. Note the values of SCD in MERRA2 are small due to the large SD required to have full snow cover in MERRA-2, hence due to the SCF being below the 0.5 threshold used in the SCD definition (see Fig. 3).

Moving to the geographical distribution over the TP region, Fig. 5 shows maps of the 5-year mean SD in January for each of

the 5 re-analyses, with the in-situ data embedded in each map. January was chosen to illustrate a mid-winter month when snowfall is relatively common and differences between re-analyses are large. In the southeast TP, where the stations are located, only JRA-55 and ERA-I have SDs comparable to in-situ observations, as was shown in Fig. 2. In the western station-void part, there are also large differences in the snow depth (up to factor 5) among the re-analyses, especially along the arc of the Himalayas and other high mountain ranges. Figure 6 is similar to Fig. 5 but represents January anomalies from

the satellite MW data for the combined years 2009 and 2010. In the southeastern part, SD in ERA-I and JRA55 are smaller and closer to in-situ data, consistent with what was shown in Fig. 2. Figure 6 also reveals that re-analyses like ERA5 and JRA55 have an excessive SD over the high mountains of the Western Himalayas compared to satellite MW data, unlike ERA-I with its high-altitude snow cover assimilation. Figure 7 shows maps of the 5-year mean SCF in January for each of the 5 re-analyses, for the two (low and high) estimates from IMS, again with the in-situ SCF embedded in the maps. While

there is a good agreement between the in-situ data and the IMS low estimate, as shown earlier at station locations, the overestimation by ERA5 extends over the eastern and western parts of the TP. Nevertheless, some parts of Central Tibet and the Taklamakan desert to the North, are snow-free in January in both IMS and most re-analyses. ERA-I agrees much better with the IMS data, while JRA-55 SCF is also much too high, due to the SD vs. SCF conversion as explained above. On the contrary, the MERRA-2 SCF is exceedingly low and featureless.


## 4 Discussion

A first important issue to mention is that the conversion of SD to SCF differs significantly among re-analyses. SCF is perhaps the most important snow-related climate driver for the surface budget, since the short-wave snow albedo feedback is strong on the TP, especially in the spring. The snowpack is also quite thin, so that thermodynamical feedbacks linked to





snow thickness are less important. A 100 % snow cover might correspond to snow depths ranging from 26 cm to 2 cm, depending on the re-analysis considered. There may also be some dependence on snow density as in the cases of ERA-I and MERRA-2. Hence, while JRA-55 has the best performance among re-analyses for snow depth, the snow cover is exaggerated due to the conversion to 100 % SCF obtained with a small SD threshold value of 2 cm. While MERRA-2 SCF

compares best with station data among the among re-analyses (see Fig. 4, right), being close to the low IMS estimate, it is exceedingly low over the TP overall (Fig. 7), given the comparatively high value (26 kg m$^{-2}$) used for snow mass to get a fully covered snow area (Reichle et al. 2017a).

The high snow bias in re-analyses might be partly due to a missing snow removal process. Gordon et al. (2006) reported that blowing snow events are common in the Canadian Prairies and in the Arctic. As snow is carried in suspension, it may

sublimate. In a model study, Pomeroy and Li (2000) predicted that two-thirds of the snow transported from the surface to the atmosphere is sublimated in the Prairie regions, and one half in the Arctic regions. Brun et al. (2013), also using Gordon's parameterization, argued that this was an important process in simulating snow dynamics in Northern Eurasia. Processes like the blowing of snow and the sublimation of the blown snow might be important processes in the windy and dry conditions of the TP.  To test the hypothesis that sublimation of blowing snow is an important missing process, we performed a special

experiment with the ERA5 land surface model, where a simple parametrisation of blown snow sublimation has been introduced, as described in the Appendix. Other processes such as the impacts of blown snow on surface roughness or of snow drifts were not tested. Station-averaged SD and SCF from this experiment (called Offline-ERA5L-EXP) are shown as dashed lines on Figs. 2d and 3d, respectively, and as maps of SD and SCF on Figs. 5, 6 and 7. With the inclusion of snow sublimation, SDs decrease significantly during both the snow accumulation and snow melting seasons, and the seasonal SD

maximum is also lower. While Figs. 2d and 3d indicate that the improved land re-analysis (Offline-ERA5L-EXP) remains positively biased compared to station data, both in terms of snow depth and snow cover, Fig. 4 nevertheless shows a clear drop in RMSE. Table 2 also indicates an increase in the daily temporal SD correlation when sublimation is accounted for.

However, we cannot conclude that blowing snow sublimation is the main reason for the large biases in SD or SCF, caused in all likelihood by excessive snowfall, a bias that is common among many climate and forecast models for this region. In order

to demonstrate this, we first present in Fig. 8 the 5-year mean seasonal cycles of minimum temperature, maximum temperature, total precipitation rate (i.e. liquid and snowy precipitation), snow cover and snow depth, based on monthly means and averaged over all the 33 stations. Figure 8 reveals that all re-analyses have a cold temperature bias compared to in-situ observations, esp. in maximum temperature, which is largely consistent with their respective high snow bias. For example, ERA-I is warmer and closer to the in-situ observations than ERA5, likely due the latter having a higher amount of

snow on the ground. The good performance of ERA-I (and also of the first-generation MERRA) against in-situ station temperature data had been noted by Wang and Zeng (2012). All re-analyses have a precipitation excess (Fig. 8c), except MERRA-2 which uses observed precipitation data, including over the TP region. Incidently, the excellent performance of MERRA-2 in terms of precipitation does not however make MERRA-2 mean snow depth over the stations better than the other re-analyses (see Fig. 4, left).  Palazzi et al. (2013) also reported the precipitation excess in ERA-I, for example. In fact,





ERA-I displays a precipitation excess greater than the more recent ERA5, despite having smaller SDs. The improvement in precipitation in ERA5 is likely due to its higher horizontal resolution, allowing better representation of synoptic events. Note that the precipitation seasonal cycle is largely dominated by summer monsoonal precipitation in the southeast TP. While our focus here is on the cold season, when precipitation is much smaller but likely falling as snow, there is still a positive bias in

the re-analyses then, indicating excessive snowfall. In fact, the largest relative climate model bias for precipitation over land, globally, is on the TP (Flato et al., 2013). Hence, a remaining question that must be addressed in future studies is the origin of the seasonal moisture transport to the TP region leading to the excess precipitation in models. We note that there is a strong regional seasonality in the precipitation pattern over the TP. The wet season precipitation (May-September) accounts for more than 70 % of the total annual precipitation over the south and southeast TP (where stations are located), a region

that falls under the influence of the Asian summer monsoon (Maussion et al., 2014). Over the western and northwest parts of the TP on the other hand, the wet season precipitation is less than 300mm. This region is influenced by strong winter and spring westerlies and transient migratory synoptic eddies embedded in them, the so-called *westerly disturbances*, which provide a significant portion of the annual mean precipitation, esp. over the Western Himalayas (Tiwari et al., 2017). An effective barrier effect, inhibiting large-scale moisture transport over the region, is not captured by any of the models

underpinning the re-analyses used here. The impact of having higher model horizontal resolution to resolve the complex topography has to be considered. Lin et al. (2018) carried out simulations for the summer monsoon season with the weather research forecasting model (WRF) at resolutions of 30, 10 and 2 km, and found that the finest resolution (2 km) diminished the water vapour transport to the TP, and the precipitation bias. They also noticed a large improvement when model resolution changed from 30 to 10 km. The insufficient subscale orography variance and orographic drag seems to be a key

factor (Zhou et al., 2019). Based on these results, it appears that the highest resolution in atmospheric re-analyses examined here (near 32 km in ERA5) remains insufficient, as demonstrated by the precipitation excess compared to station data in Fig. 8c.

## 5 Summary

We have shown that several snow re-analyses used for forecast model initialisation produce an over-extensive snowpack in

winter and spring over parts of the TP. This is at odds with observational studies revealing that snowfall events are very transient, that the snow cover vanishes rapidly on time scales of days, and that large parts of the TP can remain snow-free in winter (e.g., Basang et al., 2017; Li et al., 2018). The reanalyses that assimilate local in-situ observations or IMS snow-cover are better capable of representing the shallow, transient snowpack over the TP region. JRA-55 has the best performance among re-analyses for snow depth. Considering the family of re-analyses from the ECMWF, we attribute with confidence

that the underperformance of ERA5 in terms of SD compared to its older counterpart ERA-I, is due to the lack of IMS data assimilation at high altitudes, incl. over the TP.

The inclusion of snow sublimation hence constitutes a significant improvement and validates the postulated hypothesis that it is an important missing process in the ECMWF land surface model. Due to the difficulties in obtaining high quality

observations relevant to process studies (i.e., snowfall and sublimation) in the harsh TP environment, our results provide some guidance on potential enhancements of surface processes in future reanalyses.

Pending a solution for the common model precipitation bias over the Himalayas and TP, which may require much higher horizontal resolutions than currently used in global re-analyses, improved snow initialisation through better use of observations in the analyses may improve the medium-range or seasonal forecasts. This will be the subject of a follow-up publication.

**Acknowledgment**. We acknowledge the support of the International Space Science Institute in Beijing, through the working team "Snow reanalyses over the Himalaya-Tibetan Plateau region and the monsoons" over the years 2016-2018 (Team leaders: Yvan Orsolini and Gianpaolo Balsamo). Emanuel  Dutra was supported by the Portuguese Science Foundation (FCT) under project IF/00817/2015. Joaquín Muñoz Sabater is acknowledged for valuable comments on the Offline-ERA5L testing.

**Author contribution**. All co-authors participated to the analysis through the aforementioned working team. M. Wegmann and E. Dutra  drafted the figures. B. Liu and C. Zhu provided the station data.

**Data availability**. All re-analyses data are publically available: MERRA-2 from NASA Goddard Earth Sciences Data and Information Services Center, ERA5/ERA-I/ERA-land from the European Centre for Medium-Range Weather Forecasts (ECMWF), and JRA-55 from the Japanese Meteorological Agency. The station data has been made available through the Chinese Meteorological Administration (CMA). IMS snow cover is publically available from NOAA, and the long-term MW satellite data is available from CAREERI.

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

## Appendix A

### 1. ERA-Interim

Snow is represented as a single bulk layer on top of the soil column with prognostic temperature, mass, density and albedo. Snow density is constant with depth, increasing exponentially with snow age, and decreasing after snowfall events assuming a constant snowfall density of 100 kg m$^{-3}$. Snow albedo reduces exponentially as snow ages over low vegetation or bare soil, but is constant in time for snow under high vegetation, and is reset to a maximum value of 0.85 after snowfall event.

A snow depth analysis is performed using a Cressman (1959) interpolation with successive corrections. Station observations of snow depth are used, however there are no stations used over the TP. Gridded snow cover from IMS is also assimilated since 2004, using the coarser 24km resolution product as detailed in Drusch et al. (2004). The snow depth analysis is relaxed toward a climatology when observations are unavailable. Snow cover fraction is a diagnostic variable derived from the relation

SCF = min(1, SWE/15)



where SWE is the snow mass (kg m$^{-2}$). A layer of 15 kg m$^{-2}$ represents 100 % snow cover (15 cm depth considering a snow density of 100 kg m$^{-3}$, or 5 cm depth if considering a snow density of 300 kg m$^{-3}$).

## 2. ERA5

The snow model is also a bulk single layer as in ERA-I, with the same prognostic variables but with several modifications, in particular a diagnostics of snow liquid water content, a revised snow cover fraction formulation and snow density evolution (Dutra et al., 2010). A two-dimensional optimal interpolation snow analysis, including the water equivalent, temperature, and density of snow, is performed (de Rosnay et al., 2014), using station observations of snow depth and the gridded snow cover fraction product from IMS 4km resolution product (de Rosnay et al., 2015). Unlike ERA-I, IMS data is not used above

1500 m, i.e. in high altitude regions, which includes the TP. No station data over the TP is used. Unlike ERA-I, the snow depth analysis is not relaxed toward a climatology when observations are unavailable. Snow cover fraction is a diagnostic variable derived from the relation

SCF = min(1, SD/10)

where SD is the snow depth in cm so that a layer of 10 cm represents 100 % snow cover.

## 3. Offline-ERA5L-CTLR and Offline-ERA5L-EXP

We use two additional off-line simulations of the HTESSEL model forced by meteorological data from ERA5. The land model is hence the same as used in ERA5 but there is no assimilation of snow or land observations, contrarily to the ERA5 re-analyses. As ERA5-land is not yet officially released and may be carried with a slightly different land model version, we

call this off-line run ERA5-land-CTLR. In addition, a simple parametrization of snow sublimation due to blowing snow, as proposed by Gordon et al. (2006) and used in Brun et al. (2013), is newly implemented in HTESSEL. The sublimation of blowing snow (Qs, kg m$^{-2}$ s$^{-1}$) is computed as in Gordon et al. (2006):

$$Q_s = A \left(\frac{T_0}{T_a}\right)^{\gamma} U_t \rho_a q_{sa} (1 - Rh_a) \left(\frac{U_a}{U_t}\right)^{B}, U > U_t$$

where A and B are dimensionless constants ( A=0.0018, B=3.6), $\gamma$=4, $T_0$ is the water freezing temperature. The formulation

requires the lowest model level (at 10 m height) fields of air temperature ($T_a$), air density ($\rho_a$), wind speed ($U_a$), relative humidity ($Rh_a$) and saturation specific humidity ($q_{sa}$). $U_t$ (m s$^{-1}$) is the threshold for initiation of blowing snow also used by Gordon et al (2006) following Li and Pomeroy (1997)

$U_t = 6.98 + 0.0033(T_a - 245.88)^2$

This simple approximation only represents the snow mass removal by sublimation due to the blowing of snow, but doesn't

account for the change in the energy budget. Hence, the energy required to sublimate the snow is not taken from the surface or the lower atmosphere. While this approximation would not be valid in a coupled land-atmosphere simulation, it seems appropriate in an off-line simulation. We call this new experiment ERA5-land-EXP.



## 4. JRA-55

The land surface model in JRA-55 is the Simple Biosphere model (SiB) (Sellers et al. 1986) which represents snow mass on the ground and its evolution by updating several parameters and calculations. A separate optimal interpolation-based snow depth analysis is performed once per day (at 18UTC). The first-guess background state combines the land-surface analysis and Special Sensor Microwave/ Imager (SSM/I) and the Special Sensor Microwave Imager Sounder (SSMIS) snow depth satellite observations. The analysis also considers in-situ observations of snow depth over China from CMA archives, including several stations over the TP (Onogi et al., 2007; Kobayashi et al., 2015). Snow cover fraction is a diagnostic variable derived from the relation

$$SCF = (\min(1,(SD)/2)$$

where SD is the snow depth in cm, so that a layer of 2 cm represents 100 % snow cover.

## 5. MERRA-2

The evolution of snow mass, depth, and heat content is simulated within each catchment using a three-layer snow model (Reichle, et al., 2017b). Density in each layer evolves via parameterized representations of compaction due to snow overburden and melting/refreezing (Stieglitz et al. 2001). Snow is redistributed among layers as necessary to keep the surface layer shallow enough to respond to diurnal variability. The albedo of snow-covered land surface depends on snow density and vegetation type.

No snow data assimilation is performed. However, MERRA-2 performs an on-line correction for precipitation using two different, gridded precipitation datasets from NOAA Climate Prediction Center, namely the Unified Gauge-Based Analysis of Global Daily Precipitation (CPCU) product and the Merged Analysis of Precipitation (CMAP) product (Reichle et al., 2017a,b). Snow cover fraction is a diagnostic variable derived from the relation

$$SCF = \min(1,SWE/26)$$

where SWE is the snow mass in kg m$^{-2}$.



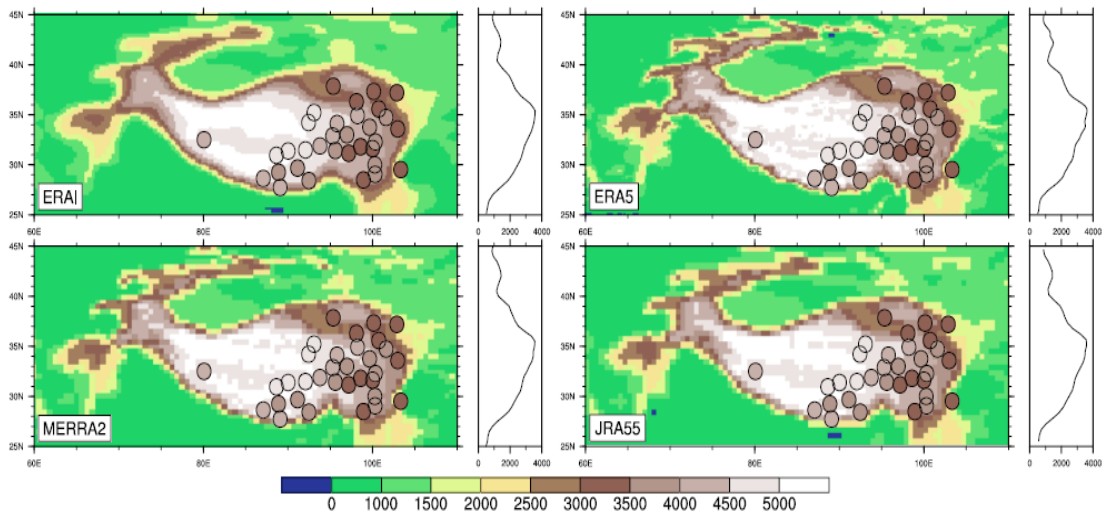

**Figure 1: Topographic maps of the TP region (altitude in m) at the horizontal resolution of each re-atmospheric analysis. The locations of the 33 stations are also shown with the coloured circles indicating the altitude range. The longitudinal average of the altitude over the represented sector (60° E-110° E) is shown to the right of each map.**



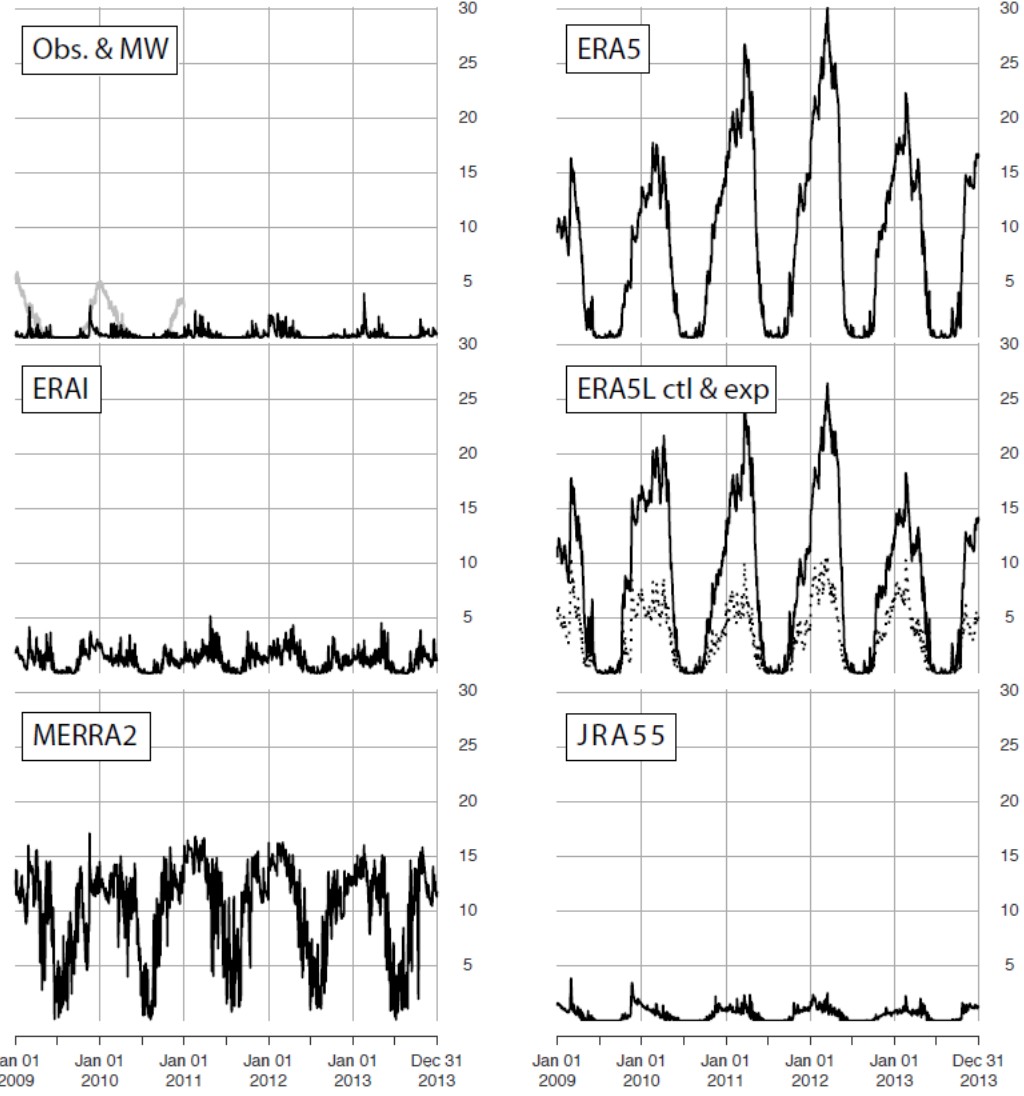

**Figure 2: Comparison over the 5-year period 2009-2013 of daily SD [in cm] between the observations, comprising both in-situ station (full black line) and MW data (grey line over 2009 and 2010), and the 5 re-analyses collocated at the station coordinates, namely ERA5, ERA-I, Offline-ERA5L-CTLR (full line) and Offline-ERA5L-EXP (dash line), MERRA-2 and JRA-55. The average is carried out over the 33 stations shown in Fig. 1.**



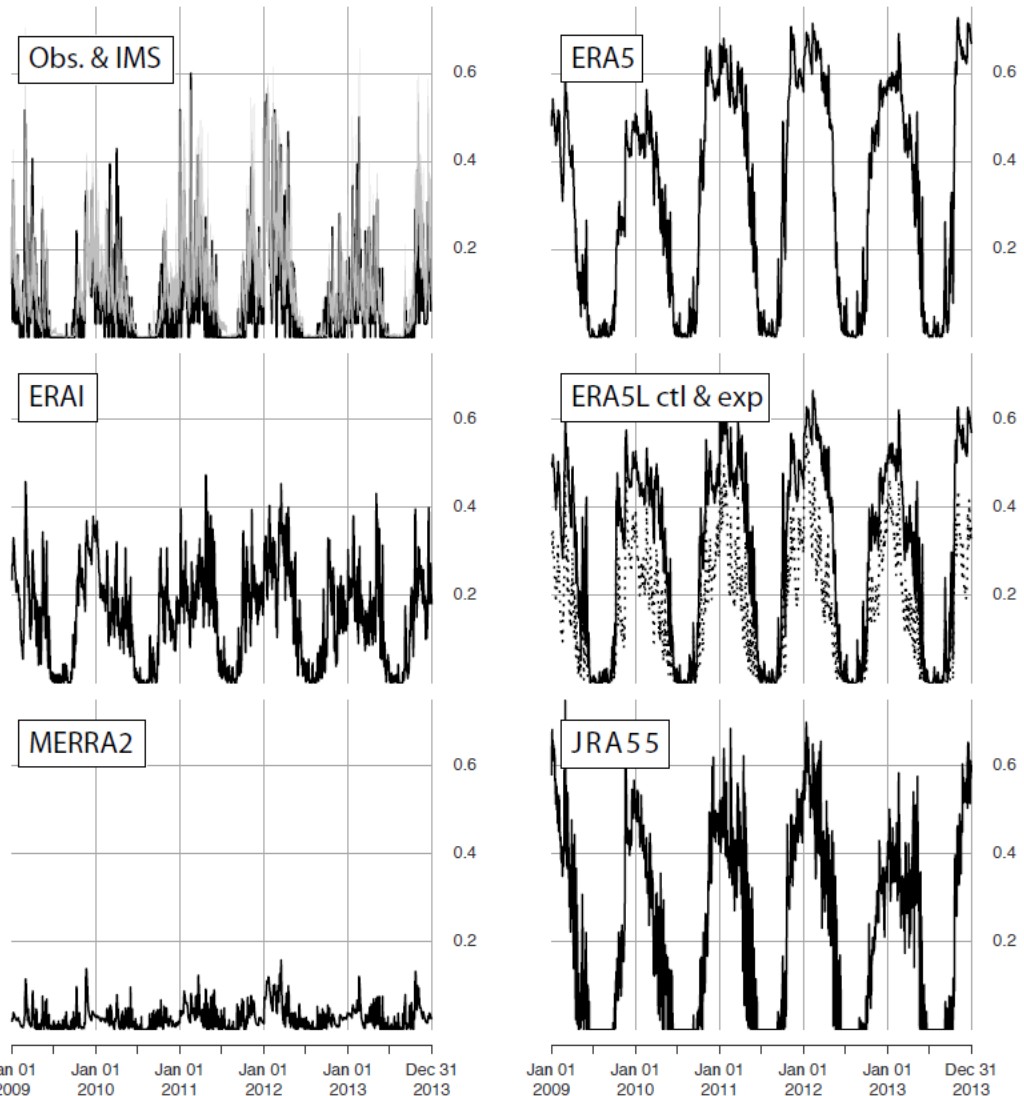

Figure 3: Comparison of daily SCF [between 0 and 1] over the 5-year period 2009-2013 between the observations, comprising both the in-situ station and IMS satellite data, with the 5 re-analyses analyses collocated at the station coordinates, namely ERA5, ERA-I, Offline-ERA5L-CTLR and Offline-ERA5L-EXP, MERRA-2 and JRA-55. The data has been averaged over the 33 stations. The IMS data is represented by the range (grey shading) between the low and high estimates, resulting from converting the binary 4-km data to a grid comparable to the re-analyses resolution.

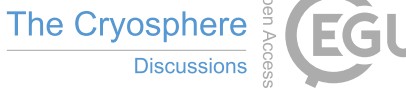

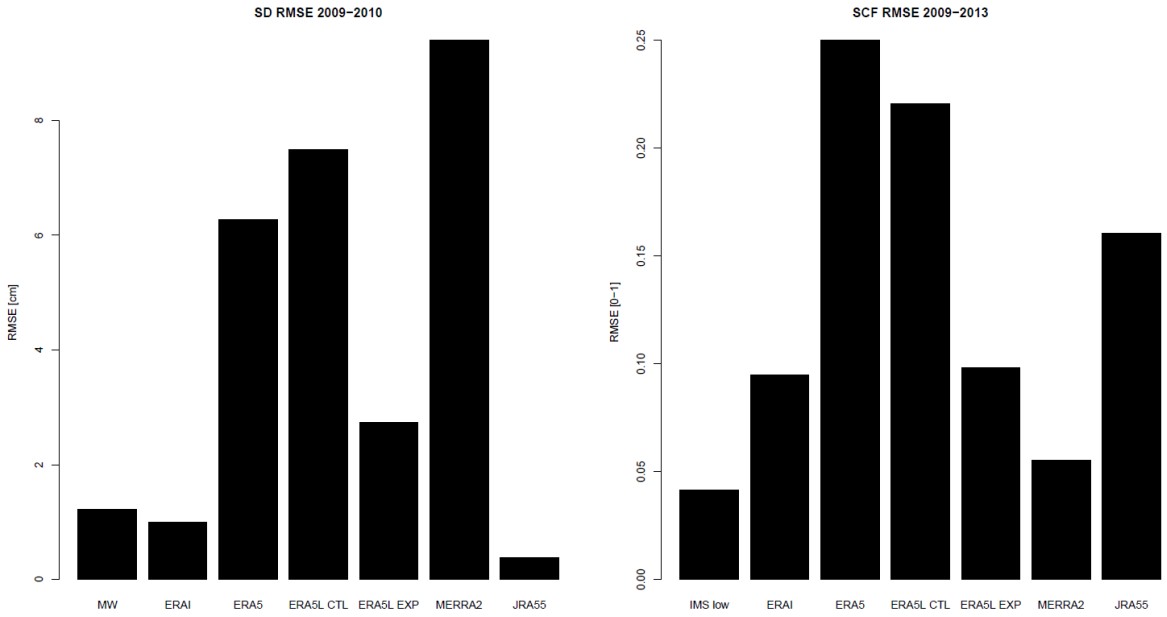

**Figure 4: Estimate of the root-mean-square error (RMSE) between daily station data and collocated re-analyses or satellite observations for SD [in cm] (left panel) over the two years when satellite MW data is available (2009 and 2010), and for SCF [between 0 and 1] (right panel), over the 5-year period (2009-2013).**




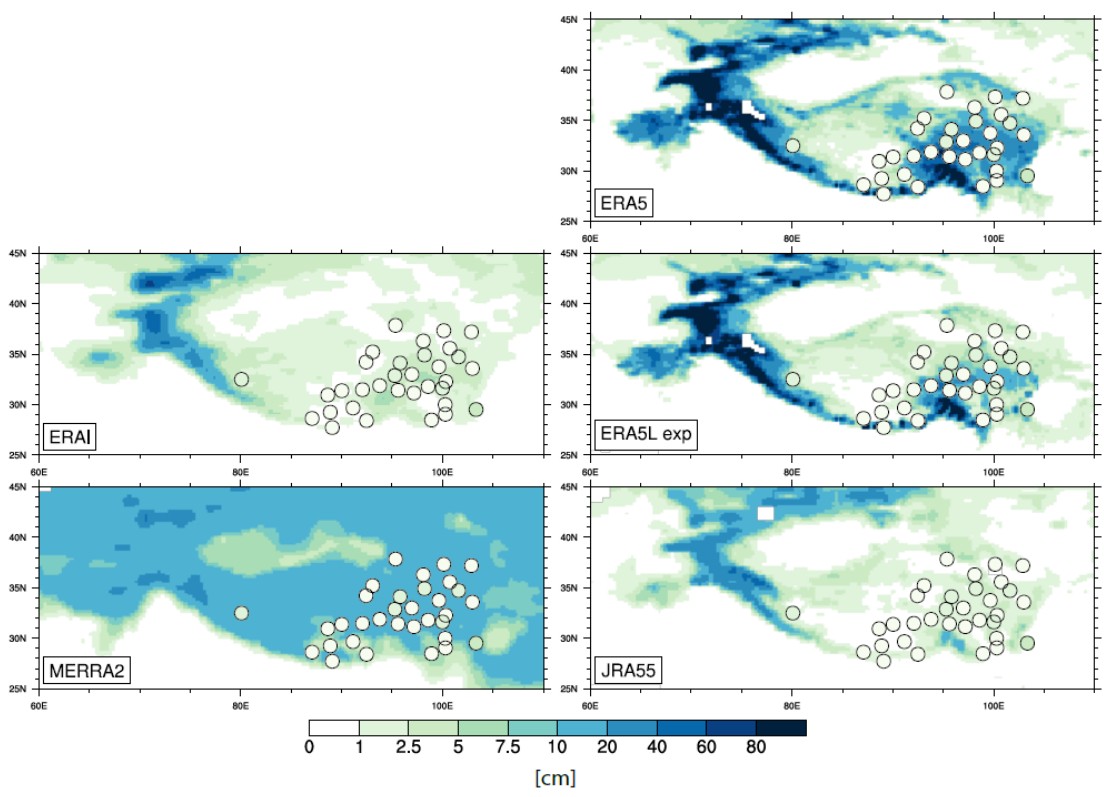

**Figure 5: Maps of the 5-year mean SD [in cm] over the TP region in January for each of the 5 re-analyses (ERA5, ERA-I, Offline-ERA5L-EXP, MERRA-2 and JRA-55) with the in-situ SD [in cm] embedded locally in each map.**





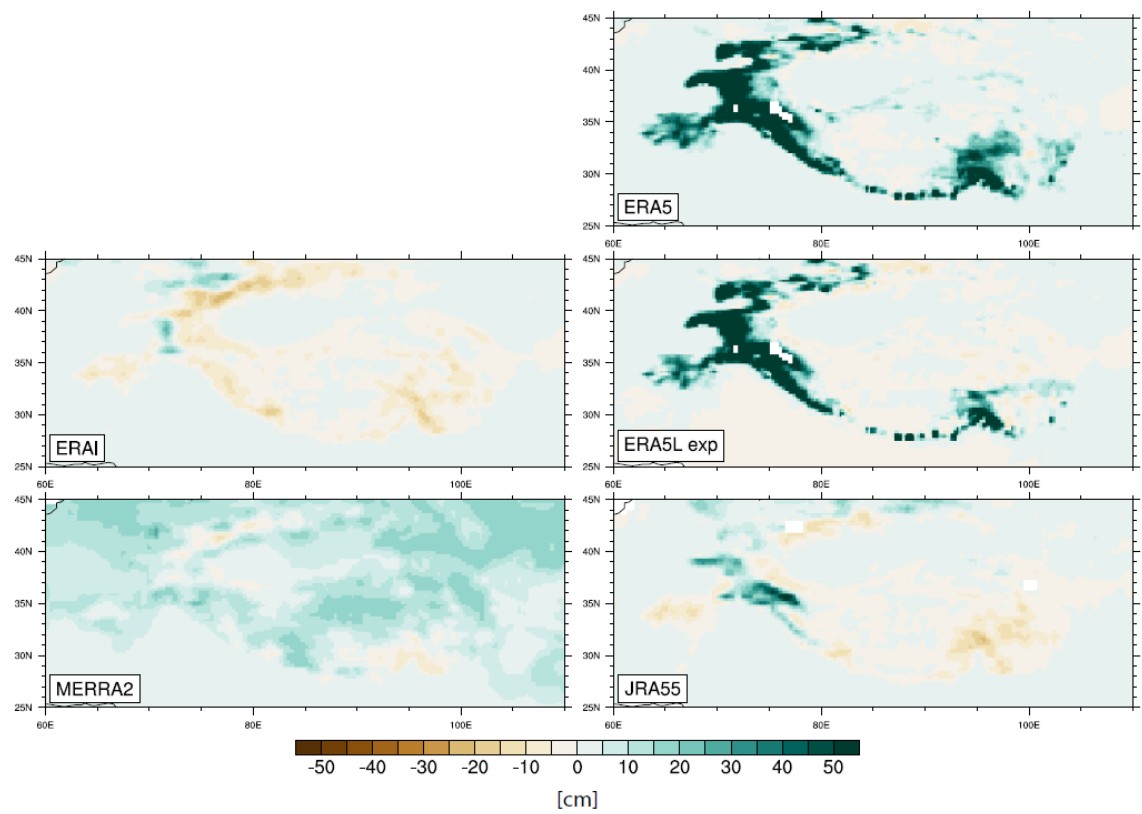

**Figure 6: Maps of the 2-year mean SD January anomalies [in cm] from the satellite MW data for the years 2009 and 2010 over the TP region for each of the 5 re-analyses (ERA5, ERA-I, Offline-ERA5L-EXP, MERRA-2 and JRA-55).**





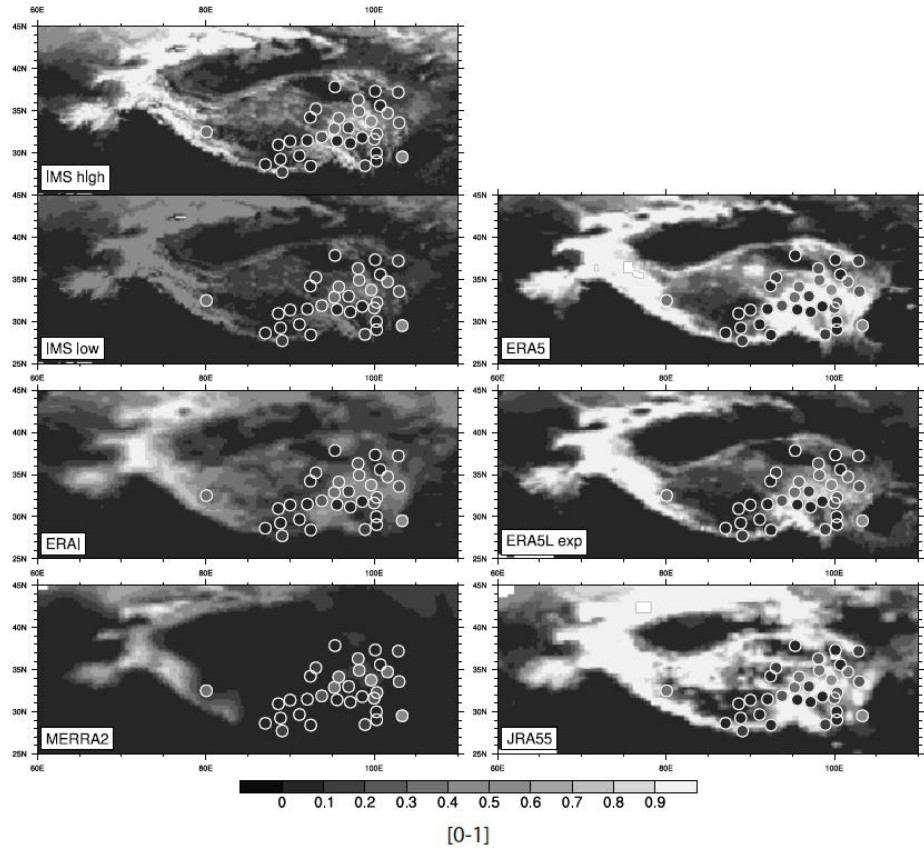

**Figure 7: Maps of the 5-year mean SCF [between 0 and 1] over the TP region in January for the IMS satellite data (two maps for the high and low estimates) and for each of the 5 re-analyses (ERA5, ERA-I, Offline-ERA5L-EXP, MERRA-2 and JRA-55), with the in-situ SCF observations [between 0 and 1] embedded locally in each map.**



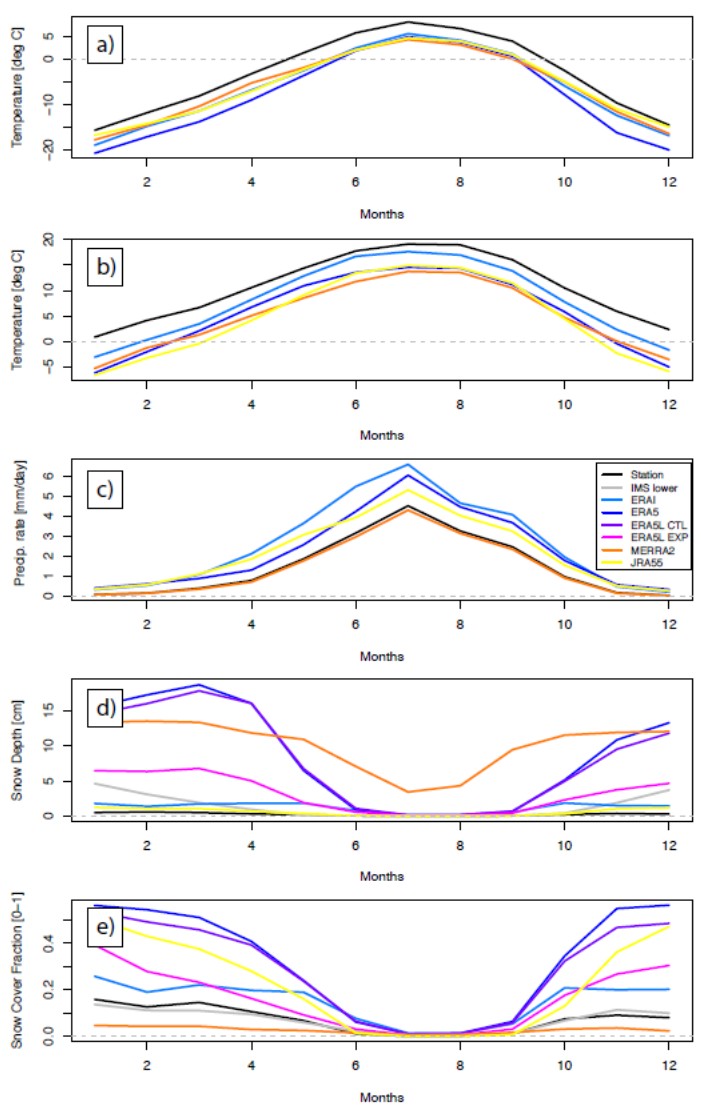

**Figure 8: The 5-year mean (2009-2013) monthly-mean seasonal cycle for minimum temperature [in deg C] (a), maximum temperature [in deg C] (b), precipitation rate [in mm per day] (c), SD [in cm] (d) and SCF [between 0 and 1] (e), averaged over all the 33 stations.**



**Table 1**: General characteristics of the re-analyses

|  | **ERA-Int** | **ERA5** | **Offline-ERA5L-CTRL** | **MERRA-2** | **JRA-55** |
|---|---|---|---|---|---|
| Approximate Spatial resolution | ~79km | ~32km | ~9km | ~50km | ~55km |
| Land Model version | TESSEL | HTESSEL | HTESSEL-CY43R1 | Catchment LSM | SIB |
| Atmospheric Model | IFS Cy31r2 | IFS Cy41r2 | NONE | GEOS 5.12.4 | JMA GSM |
| Vertical levels | 60 levels, up to 0.1 hPa | 137 levels, up to 0.01 hPa | NONE | 72 levels, up to 0.01 hPa | 60 levels, up to 0.1 hPa |
| Snow model | 1-layer | 1-layer | 1-layer | 3-layer | 1-layer |



| Snow depth | IN-SITU | MW | MERRA2 | JRA55 | ERA-I | ERA5 | ERA5landCTRL | EXP |
|---|---|---|---|---|---|---|---|---|
| IN-SITU | 1 | | | | | | | |
| MW | 0.32 | 1 | | | | | | |
| MERRA2 | 0.49 | 0.52 | 1 | | | | | |
| JRA55 | 0.72 | 0.81 | 0.65 | 1 | | | | |
| ERA-I | 0.53 | 0.48 | 0.71 | 0.6 | 1 | | | |
| ERA5 | 0.51 | 0.76 | 0.72 | 0.75 | 0.57 | 1 | | |
| ERA5landCTRL | 0.52 | 0.71 | 0.71 | 0.77 | 0.62 | 0.96 | 1 | |
| ERA5landEXP | 0.63 | 0.74 | 0.74 | 0.81 | 0.64 | 0.91 | 0.94 | 1 |

| Snow cover | IN-SITU | IMS | MERRA2 | JRA55 | ERA-I | ERA5 | ERA5L-CTRL | ERA5L-EXP |
|---|---|---|---|---|---|---|---|---|
| In-SITU | 1 | | | | | | | |
| IMS | 0.78 | 1 | | | | | | |
| MERRA2 | 0.63 | 0.78 | 1 | | | | | |
| JRA55 | 0.73 | 0.76 | 0.57 | 1 | | | | |
| ERA-I | 0.59 | 0.74 | 0.7 | 0.67 | 1 | | | |
| ERA5 | 0.58 | 0.8 | 0.89 | 0.6 | 0.77 | 1 | | |
| ERA5L-CTRL | 0.6 | 0.82 | 0.89 | 0.63 | 0.82 | 0.97 | 1 | |
| ERA5L-EXP | 0.68 | 0.83 | 0.86 | 0.72 | 0.77 | 0.91 | 0.93 | 1 |

**Table 2: Temporal correlation matrix between in-situ observations and re-analyses or satellite data, for daily SD (top) and daily SCF (bottom). The IMS low-estimate has been used. Correlations are calculated over the whole 5-year period, except for MW SD data when the two years 2009 and 2010 are used, and averaged over the 33 stations.**



|  | 5-year mean | 2009 | 2010 | 2011 | 2012 | 2013 |
|---|---|---|---|---|---|---|
| **In-SITU** | 3.08 | 3.06 | 2.31 | 3.28 | 3.81 | 2.92 |
| **IMS** | 3.20 | 2.58 | 1.90 | 4.26 | 3.80 | 3.47 |
| **MERRA2** | 0.03 | 0.06 | 0.02 | 0.015 | 0.05 | 0.02 |
| **JRA55** | 7.31 | 7.28 | 6.24 | 8.21 | 7.30 | 7.50 |
| **ERA-I** | 2.95 | 3.16 | 2.20 | 3.38 | 3.31 | 2.73 |
| **ERA5** | 8.99 | 4.92 | 8.77 | 10.70 | 10.44 | 10.14 |

|  | IN-SITU | IMS | MERRA2 | JRA55 | ERA-I | ERA5 |
|---|---|---|---|---|---|---|
| **IN-SITU** | 1 |  |  |  |  |  |
| **IMS** | 0.88 | 1 |  |  |  |  |
| **MERRA2** | 0.29 | 0.15 | 1 |  |  |  |
| **JRA55** | 0.81 | 0.86 | 0.18 | 1 |  |  |
| **ERA-I** | 0.84 | 0.79 | 0.34 | 0.72 | 1 |  |
| **ERA5** | 0.78 | 0.82 | 0.11 | 0.86 | 0.73 | 1 |

**Table 3: Values of the annual means of monthly snow cover days averaged during the whole 5-year period (the 2nd column) and during 2009 to 2013 (the 3rd to 7th columns), respectively (top), for each dataset. Temporal correlation matrix between in-situ observations and re-analyses for monthly SCD during the whole 5-year period (bottom). The SCDs are defined as the days when the SCF is greater than 0.5, and are averaged over the 33 stations.**