# Peer review of "Evaluation of snow depth and snow-cover over the Tibetan Plateau in global reanalyses using in-situ and satellite remote sensing observations"

_The Cryosphere, 2019_

## Referee Comment (RC1) · Anonymous Referee #1 · 8 Apr 2019

General: The authors have presented results on intercomparing snow depth (SD) and snow cover fraction (SCF) estimates in different reanalyses and observational data sources over the TP. This is an important topic with relatively rare previously documented literature, due to the difficulty in understanding snow conditions over the TP using both observational and modeling techniques, as well as the importance of TP snow to climate prediction. In addition to data intercomparison, authors also implemented a simple parameterization of blown snow sublimation in the ERA5 LSM to further show some problems are caused by blowing snow events previously missing in models, which offers potential explanations on the snow positive bias problem. While I found this paper interesting to the readership of The Cryosphere, below are some

specific comments for the authors to address before it can move further.

1. Around P5L10: change "didn't" to "did not"

2. P5L10: it is a bit ambiguous "they incorporate snow observations over the TP (ERA-I, JRA-55)". (1) Do they incorporate TP-exclusive observations, or do they incorporate global observations that covers the TP? (2) Any reference?

3. Around P5L10: why 1cm is used for 100% SCF? Any reference for that? Would 1cm be too small of a value for 100% SCF?

4. Major-ish: P6L10: what type of uncertainties? How would they potentially impact the results? Authors need to be more specific about this.

5. Around P6L15: change to "IMS data is not used above 1500 m in ERA-5, i.e., xxx, while it was still used in ERA-I".

6. Around P6L20: change "specificity" to "special conditions"

7. Around P6L25: change to "a horizontal resolution of 0.25 degree" or "a grid cell size of 0.25 by 0.25 degree"

8. Around P7L10: what does "on a non-zero accumulation throughout winter" mean?

9. P7L10: change "by a factor 5 to 10" to "by a factor of 5 to 10"

10. P7L12: "remaining SD" does not make sense? Consider revise it

11. In presenting the SD evaluation results: instead of mentioning the MW data as: "The temporal correlation with the in-situ data is poorer than for the re-analyses though (0.32), but is established over two years only.", I think authors need to more objectively state that the MW data has some problem capturing the SD variability. To me, this level of low correlation (even only with two years of data) already can help make this conclusion

12. P7L18: what does "the MW data is able to represent shallow layers of the order of

5 cm or less."? Authors need to revise it with a clearer statement, e.g., "the MW data is able to estimate the SD as being smaller than 5 cm, which is significantly closer to the in-situ observations than re-analyses other than ERA-I"

13. P7L23: Is IMS SCF assimilation the only difference between ERA-I and ERA5? If not, how can the conclusion be reached on "The tendency to reduce or remove the snowpack provided by the IMS observational constraint during assimilation appears to play a major role in bringing ERA-I SDs significantly closer to the in-situ observations."? This seems to be a conjecture not well supported.

14. P8L5: add "(see Appendix)" after "a thin 2 cm layer is equivalent to 100% SCF"

15. Fig. 3: the grey shading for MW data cannot be clearly seen. Can the opacity be changed lower and the shading be put behind the black line?

16. P8L12: add "(see Appendix)" after referring to MERRA-2's high threshold. In addition, can the authors provide an estimate of threshold value in SD, using the 26 SWE threshold and an estimate of snow density? This way, readers will be better informed about the threshold problems used in different reanalyses in a consistent way.

17. The Discussion part reads very interesting and reasonable to pinpoint to possible reasons for reanalyses bias problems. Suggestions on P9L20: add some references after "... for this region". For example, Su et al. (2013; JC) suggested possible snow problems over the TP, which may explain the GCM cold biases there: https://journals.ametsoc.org/doi/10.1175/JCLI-D-12-00321.1.

18. P10L31: change "incl." to "including"

19. Major-ish: About the parameterization of blowing snow sublimation scheme: while I think results in this section are interesting and provide possible reasons on the snow bias problem, the three parameters (A, B, and gamma) used in the parameterization were derived from Gordon et al. (2006) as best fit to their locations, and hence are

highly empirical. How would they potentially influence the results? What if the positive snow bias in TP is not caused by the unaccounted blowing snow sublimation, but it only saw reduced error because of the parameters used to compensate other sources of errors? It will be good for the authors to provide some discussions or sensitivity analyses on these issues. This is a major conclusion of this paper so it may warrant more scrutiny.

20. Major-ish: Figure 8: this part of discussion is interesting. But should SCF a more related variable to Tmax than SD? I saw SD biases are more consistently aligned with Tmin&Tmax biases (Fig. 8d). But in Fig. 8e, the SCF biases for MERRA-2 seem not supporting the Tmax cold bias. Any explanations on that?

21. Major-ish: Table 1: I see less of a need to mention atmospheric model layers/resolutions, than to mention what type of observational data were assimilated? The authors provided some information in the main texts, but not in the table. I believe such information needs to be more clearly summarized here.

22. Major-ish: for climate predictions at the seasonal time scale, the eastern TP snow will be an insignificant source of predictability due to its short memory (snow vanishes in a few days, as the authors observed). However, only one station is on western TP (where snow memory may be longer), and therefore the TP estimates accuracy over western TP is still largely unknown. How can this study inform medium- to seasonal-time scale climate predictions with the results being skewed towards those shallow snow regions? It would be good for the authors to discuss these issues.

23. The English of this paper may still benefit from corrections from native speakers.

---

## Referee Comment (RC2) · Anonymous Referee #2 · 12 Apr 2019

This is a well written concise study of Tibetan snow cover in modern reanalyses that yields clear and well justified, careful conclusions.

While some points could have been analysed or discussed in a bit more detail (for example, the altitudinal distribution of model errors with respect to observations, or the parameterized SD/SCF relationships), I do not think that this would have added much to the paper. My only complaint about the paper is that a little bit more effort could be devoted to the figures which are a bit hard to read (the color maps are not optimal, for example). At some points, the authors use abbreviations where full text would have been nicer ("incl."). In summary, I only recommend a few very minor technical

corrections and congratulate the authors on producing this excellent paper.

Some minor remarks: P3, Line 4: "the representativeness of this in-situ data" -> "these in-situ data" P3, L9: "at a high-resolution (4-km)" -> "at high resolution (4 km)" P3, L30: "Furthermore, re-analyses serve as initial conditions for prediction models, from short-term to monthly and seasonal forecasts." I think that operational analyses are used for such purposes rather than reanalyses? P4, L26: "inter-compare": compare snow cover and snow depth with what? BTW the verb inter-compare actually does not exist.

---

## Author Comment (AC1) · 12 Jun 2019

Response to the Reviewers for manuscript MS No.: tc-2019-49 (Journal: TC)
Title: Evaluation of snow depth and snow-cover over the Tibetan Plateau in global reanalyses using in-situ and satellite remote sensing observations
Author(s): Yvan Orsolini et al.

We are deeply grateful to the reviewers for their careful, in-depth reading of the manuscript.

Following the reviewer's suggestion, we started to explore the impact of varying the parameters in the parametrisation of blowing snow sublimation, based on Gordon (2006). This led us to discover a coding error that put into question the importance of blown snow sublimation. We have now made a broader sensitivity analysis with the off-line land model of ERA-5, carrying out five additional experiments, instead of a single experiment in the submitted version of the manuscript. We now vary the wind threshold in Gordon's parametrisation, but also the 100% SCF conversion threshold and, more importantly, the snowfall.

The parametrisation of the sublimation due to blowing snow involved some code change in the ECMWF ERA5-land model, but an error was introduced in the snow energy budget calculation. Due to this error, the results of the ERA5-land experiment (ERA5L-EXP) in the submitted manuscript (e.g. dashed line in Fig. 2) reflect the complete removal of snow sublimation from the energy budget, which leads a reduced cooling of the snowpack and enhanced melting. Therefore, the results presented in the manuscript about ERA5L-EXP do not reflect the role of sublimation due to blowing snow, but the role of neglecting snow sublimation altogether. After correcting the error, the effect of snow sublimation due to blowing snow using the described parameterization is negligible. The main message of the manuscript was to highlight the large discrepancies between several state-of-the-art atmospheric reanalysis in representing snow depth and cover in the Tibetan Plateau. This has not changed, and only the off-line experiment "ERA5L-EXP" was in error.  Reference to that experiment has been removed in the text and figures.

Instead, to shed some light on potential explanations to the large biases in the ERA5 reanalysis, we now provide a set of 5 additional, off-line land-only experiments (table below, and Table 3 in the revised manuscript) that test different hypotheses: (i) effect of snow sublimation due to blowing snow (BLW as in Gordon et al. 2006; and BLW_L with a lower threshold for blowing snow initiation); (ii) snow cover fraction threshold to represent 100% snow cover (changing from 10 cm to 5 cm SCF05 and 20 cm SCF20), and (iii) excessive snowfall (by reducing the snowfall by 50%).

The evaluation of the different budget terms over the 33-station mean is shown in Tables 5 and 6 of the revised manuscript, reproduced below. In particular, the effect of snow sublimation due to blowing snow has a negligible effect, but artificially reducing snowfall by 50% reduces the systematic biases significantly and increases the temporal correlation of snow depth.

| Experiment | Description |
|---|---|
| CTR | Land-only simulation with the same configuration as ERA5 in the land-surface |
| BLW | Including the effect of sublimation due to blowing snow as in Gordon et al. 2006 |
| BLW_L | As BLW, but changing the critical threshold for initiation from 6.98 to 5 m s$^{-1}$ |
| SCF05 | Using 5 cm instead of 10 cm threshold for 100% snow cover fraction |
| SCF20 | Using 20 cm instead of 10 cm threshold for 100% snow cover fraction |
| SNF50 | Reducing the snowfall rate by 50% |

| Experiment | Snow depth (cm) | Snow cover (0-1) | Sublimation (mm) | Sublimation blowing (mm) | Snow melt (mm) | Snowfall (mm) | Rainfall interception (mm) |
|---|---|---|---|---|---|---|---|
| ERA5 | 9.16 | 0.33 | - | - | - | 204.06 | - |
| ERA5L-CTRL | 8.54 | 0.29 | -37.34 | 0.00 | -184.13 | 204.06 | 17.42 |
| BLW | 8.50 | 0.29 | -36.29 | 1.17 | -183.82 | 204.06 | 17.40 |
| BLW_L | 8.39 | 0.28 | -35.17 | -3.44 | .-182.74 | 204.06 | 17.30 |
| SCF05 | 7.46 | 0.29 | -37.74 | 0.00 | -182.76 | 204.06 | 16.45 |
| SCF20 | 11.71 | 0.32 | 39.17 | 0.00 | -184.16 | 204.06 | 19.27 |
| SNF50 | 1.38 | 0.12 | -24.03 | 0.00 | 84.15 | 102.03 | 6.16 |
| OBS | 0.23 | 0.13 | | | | | |

Table 5: ERA5 and off-line ERA5-land experiments mean annual snow and associated fluxes averaged over the 33 stations. The fluxes are the annual mean accumulated: snow sublimation, snow sublimation due to blowing snow, snow melt, snowfall and rainfall intercepted in the snowpack. The last row (OBS) presents the mean snow depth derived from the station data, and the mean snow cover from the IMS satellite data (low estimate).

| Experiments | Snow depth RMSE (cm) | Snow cover RMSE (0-1) | Snow depth correlation | Snow cover correlation |
|---|---|---|---|---|
| ERA5 | 11.98 | 0.25 | 0.51 | 0.78 |
| ERA5L-CTRL | 11.19 | 0.21 | 0.50 | 0.80 |
| BLW | 11.13 | 0.20 | 0.50 | 0.80 |
| BLW_L | 10.97 | 0.20 | 0.50 | 0.80 |
| SCF05 | 9.94 | 0.20 | 0.50 | 0.81 |
| SCF20 | 14.98 | 0.25 | 0.51 | 0.77 |
| SNF50 | 1.58 | 0.08 | 0.60 | 0.79 |

Table 6. Root mean square error (RMSE) and temporal correlation for snow depth (versus stations data) and snow cover (versus IMS satellite) for ERA5 and the different experiments. The time series of the observation and experiments were first averaged over the 33 stations before the score calculations.

**Reviewer 1**

1. Around P5L10: change "didn't" to "did not"
done
2. P5L10: it is a bit ambiguous "they incorporate snow observations over the TP (ERAI, JRA-55)". (1) Do they incorporate TP-exclusive observations, or do they incorporate global observations that covers the TP? (2) Any reference?
We made the sentence more precise: "…they incorporate either local snow observations over the TP region (JRA-55) or satellite snow cover observations that encompass the TP (ERA-I, JRA-55)."

References are given in the Appendix describing the re-analyses (e.g. Kobayashi's and Onogi's papers for JRA-55)

3. Around P5L10: why 1cm is used for 100% SCF? Any reference for that? Would 1cm be too small of a value for 100% SCF?

Assuming a 100% SCF when snow depth exceeds 1 cm is according to the in-situ observing rules of the China Meteorological Administration (2003), and a reference has been given. This is a smaller threshold than used in re-analyses (where it varies between 2 and 26 cm, depending of the re-analyses). For the ERA5 system, we did a sensitivity off-line test with the land model, and changed the threshold from 10 to 5 cm. Results are shown in Tables 5-6.

4. Major-ish: P6L10: what type of uncertainties? How would they potentially impact the results? Authors need to be more specific about this.

We made the sentence more precise. "These two estimates provide a range of values, reflecting the uncertainty inherent to aggregating the 4-km binary data (e.g, a value of 1 in a pixel means 50% to 100% snow coverage)."

5. Around P6L15: change to "IMS data is not used above 1500 m in ERA-5, i.e., xxx, while it was still used in ERA-I". Done

6. Around P6L20: change "specificity" to "special conditions" Done

7. Around P6L25: change to "a horizontal resolution of 0.25 degree" or "a grid cell size of 0.25 by 0.25 degree" Done

8. Around P7L10: what does "on a non-zero accumulation throughout winter" mean?

We made the sentence more precise. "This is unlike the in-situ observations which show rapidly fluctuating snow increases with little snow accumulation throughout winter. "

9. P7L10: change "by a factor 5 to 10" to "by a factor of 5 to 10" Done

10. P7L12: "remaining SD" does not make sense? Consider revise it

Revised to : "In MERRA-2, the SD does not completely vanish in summer."

11. In presenting the SD evaluation results: instead of mentioning the MW data as: "The temporal correlation with the in-situ data is poorer than for the re-analyses though (0.32), but is established over two years only.", I think authors need to more objectively state that the MW data has some problem capturing the SD variability. To me, this level of low correlation (even only with two years of data) already can help make this conclusion

Revised to: "It also poorly captures short-term SD variability (Fig. 2) and its temporal correlation with the in-situ data is 0.32, poorer than for the re-analyses, even though calculated over two years only."

12. P7L18: what does "the MW data is able to represent shallow layers of the order of 5 cm or less."? Authors need to revise it with a clearer statement, e.g., "the MW data is able to estimate the SD as being smaller than 5 cm, which is significantly closer to the in-situ observations than re-analyses other than ERA-I"

" RMSE error is nevertheless comparable to ERA-I, and the MW data is able to estimate SD as being smaller than 5 cm, which is significantly closer to the in-situ observations than re-analyses other than ERA-I and JRA-55."

13. P7L23: Is IMS SCF assimilation the only difference between ERA-I and ERA5? If not, how can the conclusion be reached on "The tendency to reduce or remove the snowpack provided by the IMS observational constraint during assimilation appears to play a major role in bringing ERA-I SDs significantly closer to the in-situ observations."? This seems to be a conjecture not well supported.

The difference between ERA-I and ERA5 concerning snow are described in the Appendix. The high resolution of ERA5 clearly improves snow variability as indicated by the higher temporal correlation coefficient. The importance of IMS assimilation at high altitude will be supported in a follow-up

paper based on dedicated data assimilation experiments. For this paper, we have toned down our wording, indicating that we only surmise that this is the key factor.

14. P8L5: add "(see Appendix)" after "a thin 2 cm layer is equivalent to 100% SCF" Done

15. Fig. 3: the grey shading for MW data cannot be clearly seen. Can the opacity be changed lower and the shading be put behind the black line? Figures have been improved and MW data is shown separately.

16. P8L12: add "(see Appendix)" after referring to MERRA-2's high threshold. In addition, can the authors provide an estimate of threshold value in SD, using the 26 SWE threshold and an estimate of snow density? This way, readers will be better informed about the threshold problems used in different reanalyses in a consistent way. This is now done.

17. The Discussion part reads very interesting and reasonable to pinpoint to possible reasons for reanalyses bias problems. Suggestions on P9L20: add some references after ": : : for this region". For example, Su et al. (2013; JC) suggested possible snow problems over the TP, which may explain the GCM cold biases there: This new reference has been added.

18. P10L31: change "incl." to "including" Done

19. Major-ish: About the parameterization of blowing snow sublimation scheme: while I think results in this section are interesting and provide possible reasons on the snow bias problem, the three parameters (A, B, and gamma) used in the parameterization were derived from Gordon et al. (2006) as best fit to their locations, and hence are highly empirical. How would they potentially influence the results? What if the positive snow bias in TP is not caused by the unaccounted blowing snow sublimation, but it only saw reduced error because of the parameters used to compensate other sources of errors? It will be good for the authors to provide some discussions or sensitivityanalyses on these issues. This is a major conclusion of this paper so it may warrant more scrutiny.

Following that suggestion, we started to explore varying the parameters in the parametrisation based on Gordon (2006). This led us to discover a coding error that put into question the importance of blown snow sublimation. We have now made a broader sensitivity analysis with the off-line land model of ERA-5, carrying out five additional experiments, varying the wind threshold but also the 100% SCF conversion threshold and, more importantly, reducing the snowfall. (see top of rebuttal letter)

20. Major-ish: Figure 8: this part of discussion is interesting. But should SCF a more related variable to Tmax than SD? I saw SD biases are more consistently aligned with Tmin&Tmax biases (Fig. 8d). But in Fig. 8e, the SCF biases for MERRA-2 seem not supporting the Tmax cold bias. We agree that the SD biases are more aligned with temperature biases, and we changed the text to be more specific. The SCF conversion destroys the relationship, and this is clearly the case for MERRA-2, where the high threshold for 100% SCF makes it always very low. Issues with SCF conversion are in a separate section in the Discussion. The new text reads: "Figure 8 reveals that all re-analyses have a cold temperature bias compared to in-situ observations, esp. in maximum temperature, which is largely consistent with their respective thick snowpack bias. For example, ERA-I is warmer and closer to the in-situ observations than ERA5, likely due the latter having a higher snow depth."

21. Major-ish: Table 1: I see less of a need to mention atmospheric model layers/ resolutions, than to mention what type of observational data were assimilated? The authors provided some information in the main texts, but not in the table. I believe such information needs to be more clearly summarized here.
We made change to the Table 1 to incorporate information on assimilated snow data.

22. Major-ish: for climate predictions at the seasonal time scale, the eastern TP snow will be an insignificant source of predictability due to its short memory (snow vanishes

in a few days, as the authors observed). However, only one station is on western TP (where snow memory may be longer), and therefore the TP estimates accuracy over western TP is still largely unknown. How can this study inform medium- to seasonal time scale climate predictions with the results being skewed towards those shallow snow regions? It would be good for the authors to discuss these issues.

This is indeed a good point, which we now stress. The relevance of the western TP snow cover for seasonal climate, due to the limited persistence over the eastern and central TP, has indeed been emphasized by Xiao and Duan (2016; cited). The new text reads:

"Finally, we note that the particular relevance of the western TP snowpack for subseasonal-to-seasonal prediction, due to the limited snowpack persistence over the eastern and central TP, as was already emphasized by Xiao and Duan (2016). Yet, with only one station in our intercomparison, the western TP snowpack remains the less constrained by in-situ data."

**Reviewer 2**

Some minor remarks: P3, Line 4: "the representativeness of this in-situ data" -> "these in-situ data" P3, L9: "at a high-resolution (4-km)" -> "at high resolution (4 km)"  Done.
P3, L30: "Furthermore, re-analyses serve as initial conditions for prediction models, from short-term to monthly and seasonal forecasts." I think that operational analyses are used for such purposes rather than reanalyses?  The sentence has been made more precise.
"Furthermore, re-analyses are sometimes used as initial conditions for seasonal hindcasts or reforecasts."
P4, L26: "inter-compare": compare snow cover and snow depth with what? BTW the verb inter-compare actually does not exist. Done.

---

## Author Response (AR2)

Correction to the manuscript MS No.: tc-2019-49 (Journal: TC) Title: Evaluation of snow depth and snow-cover over the Tibetan Plateau in global reanalyses using in-situ and satellite remote sensing observations Author(s): Yvan Orsolini et al.

Following some interaction with scientists in charge of MERRA-2 reanalyses at an international workshop, we were made aware that, confusingly, MERRA-2 snow depth is not the gridded snow depth but rather snow depth over snow covered area. Hence, the snow cover fraction needs to be taken into account. While the MERRA-2 snow depth data has been frequently misused, we wanted to make the paper as correct as possible.

Hence, we corrected the figures in the manuscript related to MERRA-2 snow depth, namely Figures 2,4,5,6 and 8. Only the curves or maps related to MERRA-2 were changed in those figures. The correlation values related to MERRA-2 were changed in Table 2.

After that adjustment, MERRA-2 performed better than JRA-55 in terms of snow depth, and we have made minor corrections in the text to take that into account.

P6, L33: It is clearly apparent that, with the exception of MERRA-2, the re-analyses show a regular seasonal cycle, with a snowpack that grows nearly steadily during the cold season and culminates in February or March.

P7, L2: In comparison with in-situ observations, MERRA-2 has the best performance among reanalyses for both the RMSE and the temporal correlation, closely followed by JRA-55. The ERA5 reanalysis over-estimates the seasonal maximum SD by a factor of 10.

P7, L6: We removed the sentence on the summer snow excess in MERRA-2.

P7, L11 : It is not surprizing that JRA-55 performs well

P7, L30: JRA-55 is much worse than MERRA-2 for SCF, while their performance for SD was similar. P8, L14: In the southeastern part, SDs in MERRA-2, ERA-I and JRA-55 are smaller than the MW data, consistent with Fig. 2

P9,L30: Hence, while JRA-55 has excellent performance among re-analyses for SD

P10,L12: Incidentally, this may explain the excellent performance of MERRA-2 in terms of mean snow depth over the stations (see Fig. 4, left).

P11, L16: MERRA-2 and JRA-55 have the best performance among re-analyses for snow depth.

**In addition**

P11, L9: we added "and further studies on this issue are warranted.", as suggested by one reviewer.

We added Drs. Reichle and Kumar in Acknowledgements.

We corrected several typos, and moved the section header 4.3 Excessive precipitation issue, up one paragraph, since it was more suited there.